# Exclusive breastfeeding continuation and associated factors among employed women in North Ethiopia: A cross-sectional study

**Kahsu Gebrekidan**[1,2]*, **Helen Hall**[1], **Virginia Plummer**[1,3], **Ensieh Fooladi**[4]

**1** Monash Nursing and Midwifery, Monash University, Frankston, Victoria, Australia, **2** Mekelle University, College of Health Sciences, School of Nursing, Mekelle, Ethiopia, **3** Federation University Australia, School of Health, Berwick, Australia, **4** Monash Nursing and Midwifery, Monash University, Clayton, Victoria, Australia

* kahsu.gebrekidan@monash.edu

**Data Availability Statement:** All relevant data are within the paper and its Supporting information files.

## Abstract

### Background

Exclusive Breastfeeding (EBF) can prevent up to 13% of under-five mortality in developing countries. In Sub-Saharan Africa the rate of EBF at six months remains very low at 36%. Different types of factors such as maternal, family and work-related factors are responsible for the low rate of EBF among employed women. This study aimed to assess the prevalence of EBF continuation and associated factors among employed women in North Ethiopia.

### Materials and methods

A community-based, cross-sectional study was conducted in two towns of Tigray region, North Ethiopia. Employed women who had children between six months and two years were surveyed using multistage, convenience sampling. Women filled in a paper based validated questionnaire adopted from the Breastfeeding and Employment Study toolkit (BESt). The questions were grouped into four parts of sociodemographic characteristics, maternal characteristics, family support and work-related factors. Factors associated with EBF continuation as a binary outcome (yes/no) were determined using multivariable logistic regression.

### Results

Four-hundred and forty-nine women participated in this study with a mean (SD) age 30.4 (4.2) years. Two hundred and fifty-four (56.4%) participants exclusively breastfed their children for six months or more. The main reason for discontinuation of EBF was the requirement of women to return to paid employment (31.5%). Four-hundred and forty (98.2%) participants believed that breastfeeding has benefits either to the infant or to the mother. Three hundred and seventy-one (82.8%) of the participants received support from their family at home to assist with EBF, most commonly from their husbands and mothers. Having family support (adjusted odds ratio [AOR] = 2.1, 95%, CI 1.2–3.6; P = 0.005), having frequent breaks at work (AOR = 2.6, 95% CI, 1.4–4.8; P = 0.002) and the possibility of buying

**Funding:** The authors received no specific funding for this work.

**Competing interests:** The authors have declared that no competing interests exist.

or borrowing required equipment for expressing breast milk (AOR = 1.7, 95% CI, 1.0–3.0; P = 0.033) were statistically associated with an increased chance of EBF.

## Conclusion

Although returning to work was reported by the study participants as the main reason for discontinuation of EBF, families and managers' support play significant roles in EBF continuation, which in the absence of six-month's maternity leave for employed women in Ethiopia would be of benefit to both mothers and children.

## 1. Introduction

In 1990 the World Health Organization (WHO) and United Nations International Children's Emergency Fund (UNICEF) adopted a declaration on protection, promotion and support of breastfeeding, focused on the importance of exclusive breastfeeding (EBF) for at least six months [1]. Appropriate EBF practice can prevent up to 13% of under-five mortality in developing countries [2]. In light of this trend, the UNICEF and WHO set a target to increase the rate of EBF to 50% by 2025 [3, 4]. However, the rate of EBF for six months is suboptimal in many parts of the world [5]. In Sub-Saharan Africa, for example, only 36% mothers exclusively breastfeed their infants until six months [6]. In Ethiopia, the rate of EBF among employed women remains suboptimal. In Ethiopian studies conducted in Gondar town [7] and Fafan zone [2], with focus on employed and non-employed women in Ethiopia, the rate of EBF among employed women was 21% and 24.8%, respectively. Another cross-sectional study conducted in Dukem, central Ethiopia in 2015 showed that only 24.3% employed mothers exclusively breastfed their infants until six months [8].

Different factors are responsible for the low rate of EBF among employed and non-employed women. Good knowledge and positive attitude of mothers about the benefits of EBF plays an important role in its continuation until six months [9, 10]. A study in Jordan that showed support and encouragement from husbands and extended family members was associated with the increased rate of EBF for six months [11], whereas mother's return to paid employment negatively impacted the duration of EBF [7, 12, 13]. Employed women need to return to work before six months because paid employment is a necessity, not an option for many of them [14]. Studies conducted among employed women shows different work-related factors that affect the continuation EBF either positively or negatively [3, 12, 13]. Availability of physical facilities and a supportive work environment such as flexibility and having supportive managers encourages women to continue EBF [3, 15].

Kebede T. et al reported several factors that triggered EBF discontinuation including a short duration of maternal leave, full-time employment, working in private organizations, lack of flexible working hours unable to express breast milk, lack of breaks to express breast milk and the workplace being far away from the child [8]. This study provides useful insights into some of the barriers to EBF for employed women in Ethiopia; however, the findings may not be generalisable to other areas of Ethiopia, as a very diverse country with different ethnicities and cultural expectations. Hence, the results of a study conducted in central Ethiopia, for example, may not be generalizable to people who live in the northern part of the country. Moreover, maternity leave in Ethiopia was increased from three to four months in 2018. The current study aimed to assess EBF continuation and associated factors among employed

women in North Ethiopia, since the introduction of increased maternity leave. Knowledge of mothers about the benefits of EBF and its practice of EBF was also assessed.

## 2. Materials and methods

This study was part of a larger mixed-methods study looking at determinants of EBF among employed women after they returned to work. This article reports on survey results of the study. Findings are reported based on the Strengthening the Reporting of Observational Studies in Epidemiology Guidelines [16].

### 2.1 Study design and setting

This community-based, cross-sectional study was conducted in two towns in Tigray region, North Ethiopia between December 2018 and January 2019.

### 2.2 Participants

The study participants were full-time employed women who had children aged between six months and two years. Women working on contract, in casual, part-time or in their own business were excluded as they might have more flexible schedules.

### 2.3 Sampling method

Multistage sampling was used to reach the study participants. First, two zones from the seven zones in the Tigray region were selected using a convenience sampling method. The biggest town was then selected from each zone because these towns are administrative and business centres in which many employed women live. All kebeles (lowest administrative division) in each town were included in the study. The total number of participants was allocated equally for each town. Finally, purposive sampling was used to recruit women from each kebele (Fig 1).

Twenty health extension workers from both towns visited women in their home. Potential participants were invited to the study and given the explanatory statement. Those who agreed to take part in the study were provided with a hard copy, self-administered questionnaire. It was explained that the questionnaire would be collected two weeks later. The first author supervised the overall data collection process.

### 2.4 Measurement of variables

We used a survey tool adapted from the Breastfeeding and Employment Study toolkit (BESt), developed by the University of Wisconsin [17]. The tool was modified to capture local factors deemed important. In total, 11 maternal characteristics and six family support related questions were added. The questionnaire comprises a total of 66 questions, 36 of which were four-point (strongly agree, agree, disagree and strongly disagree) Likert scale questions. The questionnaire was prepared in English and translated to local language (Tigrigna). Once participants completed the questionnaire it was back translated to English for data processing and analysis. Prior to distribution, the questionnaire was piloted on five employed mothers to ensure the appropriate wording and understandability by local Tigran mothers; no amendment was necessary.

The women were asked for how long they exclusively breastfed their last child. There were five alternatives to respond to this outcome variable (not at all, three months or less, four to six months, six months and more than six months). The independent variables were grouped into four 1) demographic characteristics, age of infants and mothers, marital status, type of work,

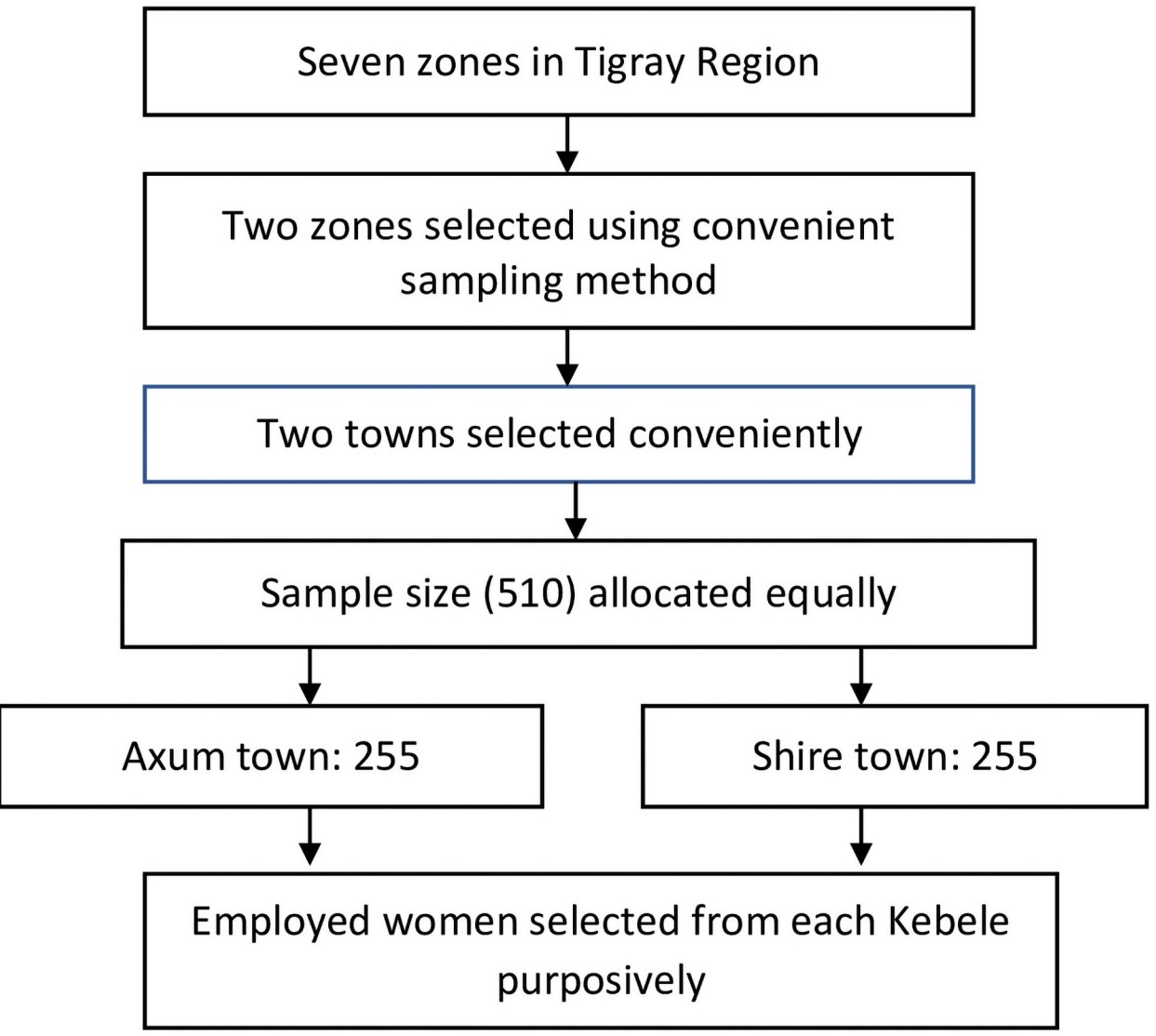

**Fig 1. Sampling method of the study.**

educational status, monthly salary of mothers, number of children, place and type of birth; 2) EBF practice and knowledge about the benefits of EBF and breast milk expression; 3) family support that focused on the support mothers obtained at home from their husband and extended family members; and 4) work-related factors. Organisational managers support, co-workers support as well as time and physical environment related questions were addressed under the work-related factors.

## 2.5 Study size

The sample size was calculated using G*power with the following parameters: a power of 0.80, true proportion of estimated prevalence of EBF among employed women (21%) based on a previous study conducted in Gondar, Ethiopia [7] and a level of significance of 0.05 and sample size was 507.

## 2.6 Statistical analysis

We summarized continuous and categorical variables using mean (± standard deviations, SDs) and frequencies (percentages), respectively. The outcome variable (duration of EBF) was recoded as either yes (breastfed for six months or more) or no (Not at all, three months or less, four to six months breastfed) for the question "did you EBF your infant for six months". The four-category responses (strongly agree, agree, disagree and strongly disagree) were collapsed into two categories (strongly agree/agree and strongly disagree/disagree).

Logistic regression analysis was used to assess the relationship between dependant and independent variables at 95% confidence interval. Initially all completed variables (n = 440) were included in the univariable logistic regression analysis. Variables with P-value <0.1 on the univariable logistic regression were included in the multivariable logistic regression. Variables with p-value 0.05 or less were considered statistically significant. SPSS software version 26 was used for the analyses.

## 2.7 Ethics approval and consent to participate

Ethical approval was obtained from Monash University Human Research Ethics Committee (ethics approval number: 13794) and Mekelle University Research Ethics Approval Committee (ethics approval number: ERC 1490/2018). Participation was voluntary and informed consent was obtained from each study participant prior to distribution of the questionnaire. To ensure their privacy, no personal identifiers of participants was used.

## 3. Results

Of the 510 questionnaires distributed, 449 (88.0%) were completed with equal number of completed questionnaires returned from each town. Nine out of 449 questionnaires were incomplete for the work-related section, and one out of nine questionnaires was incomplete for all sections except for demographic characteristics. Incomplete questionnaires were included in the descriptive statistics, but not in the logistic regression analysis.

### 3.1 Demographic characteristics of participants

The mean (SD) age of study participants was 30.4 (4.2) years. Three-hundred and seventy-nine (84.4%) participants were married and 337 (75.1%) had completed their education at diploma level or above. Three hundred and twenty-four (72.2%) respondents had either two or more live children, and 140 (31.2%) had monthly income of 107 USD or more. Four-hundred and forty-one (98.2%) mothers gave birth to their last child in a health facility and 303 (67.5%) had a spontaneous vaginal birth (Table 1).

### 3.2 Breastfeeding practices and knowledge

Four-hundred and forty-eight participants responded to the section of questionnaire investigating breast feeding practices and knowledge. Of these women, 393 (87.7%) commenced breastfeeding within one hour of birth and 254 (56.6%) mothers exclusively breastfeed their children for six months or more. For 141 (31.5%) mothers who did not EBF, their primary reason for the introduction of additional food/fluids was the requirement to return to paid employment within six months of birth.

Four-hundred and forty (98.2%) participants believed that EBF has benefits. The common reasons identified as motivation for continuing EBF included nutritional benefits (58.0%), disease prevention (66.3%) and growth and development of infants (63.8). Contraceptive effect was another benefit of EBF mentioned by 350 (78.1%) of the study participants. A total of 293

**Table 1. Demographic characteristics of study participants (n = 449).**

| Question/variable | n (%) |
|---|---|
| Age (years), mean (SD) | 30.4 (4.2) |
| Age of youngest child (months), mean (SD) | 12.1 (4.6) |
| **Marital status** | |
| Partnered | 379 (84.4) |
| Unpartnered (single, divorced, widowed) | 70 (15.6) |
| **Educational status** | |
| Secondary school or less | 112 (24.9) |
| Diploma and above | 337 (75.1) |
| **Type of work** | |
| Professional/skill | 207 (46.1) |
| Administrative | 186 (41.4) |
| Other | 56 (12.5) |
| **Monthly salary (USD)** | |
| Less than 46 | 112 (25.2) |
| 46–76 | 97 (21.6) |
| 77–107 | 100 (22.3) |
| More than 107 | 140 (31.2) |
| **Number of live children per participant** | |
| One | 125 (27.8) |
| Two or more | 324 (72.2) |
| **Place of last birth** | |
| Home | 8 (1.8) |
| Health center* | 124 (27.6) |
| Hospital | 317 (70.6) |
| **Mode of birth** | |
| Spontaneous vaginal birth | 303 (67.5) |
| Instrumental assisted vaginal birth | 48 (10.7) |
| Caesarean section | 98 (21.8) |

*Health centres are primary health care units that provide preventive and curative services with inpatient capacity of five beds.

Values are n (%) unless otherwise specified.

(65.4%) respondents reported that they had information about expressed breast milk feeding. Health extension workers and professionals were the main sources of information about expressing breast milk for 156 (34.8%) and 158 (35.3%) mothers, respectively (Table 2).

### 3.3 Family support of EBF

From the total study participants who responded (448), 371 (82.8%) reported that they received support from their family at home to continue EBF. The family members most commonly involved in supporting women to EBF were their husbands, their mothers and mothers-in-law as stated by 254 (56.7%), 172 (38.4%) and 61 (13.6%) participants, respectively. The participants also reported that 266 (59.4%) of husbands actively encouraged EBF. The common types of support women obtained at home were baby care 224 (50.0%) and staying with baby at home while they were at work 248 (55.4%). When mothers did not have support from their husband or family members, some would leave their infants with domestic workers at home 149 (33.3%) or take them to work 122 (27.2%) (Table 3).

**Table 2. Breastfeeding practices and knowledge of study participants (n = 448).**

| Question/variable | n (%) |
|---|---|
| Started breastfeeding within one hour after birth | |
| Yes | 393 (87.7) |
| No | 55 (12.3) |
| Duration of EBF | |
| Six months or more | 254 (56.6) |
| Less than six months | 194 (43.4) |
| Reasons given for not adhering to EBF | |
| Belief that breast milk alone was not enough | 38 (8.5) |
| Didn't have enough milk | 26 (5.8) |
| Started paid employment | 141 (31.5) |
| Influence from family | 3 (0.7) |
| Other | 2 (0.5) |
| Belief that EBF is beneficial | |
| Yes | 440 (98.2) |
| No | 8 (1.8) |
| Mothers' perceptions of benefits EBF for the infant** | |
| Nutritional benefits | 260 (58.0) |
| Reduces risk of some diseases | 297 (66.3) |
| Growth and development | 286 (63.8) |
| Bonding between mother & infant | 183 (40.8) |
| Mothers' perceptions of benefits of EBF for herself/women** | |
| Contraceptive use | 350 (78.1) |
| Control bleeding after birth | 115 (25.7) |
| Decrease risk of breast/cervical cancer | 112 (25.0) |
| Economic benefits | 162 (36.2) |
| Awareness of how to express breast milk | |
| Yes | 293 (65.4) |
| No | 155 (34.6) |
| Source of information about expressed breast milk feeding** | |
| Health extension workers | 156 (34.8) |
| Health professionals | 158 (35.3) |
| Mass media (radio, TV etc.) | 77 (17.2) |
| Social media | 30 (6.7) |
| Other sources* | 5 (1.1) |
| Mother fed her baby using expressed breastmilk | |
| Yes | 109 (24.3) |
| No | 339 (75.7) |
| Reason for expressed breastmilk feeding | |
| Returned to paid employment before six months | 74 (16.5) |
| Unable to breastfeed after birth | 23 (5.1) |
| Other | 12 (2.7) |

* other sources = individual woman's knowledge as a health professional, family

**possible to give more than one answer

**Table 3. Family support of EBF among study participants (n = 448).**

| Question/variable | n (%) |
|---|---|
| Family support to continue EBF following return to paid employment | |
| Yes | 371 (82.8) |
| No | 77 (17.2) |
| Members of family who provided support at home | |
| Husband | 254 (56.7) |
| Mother | 172 (38.4) |
| Mother-in-law | 61 (13.6) |
| Domestic worker | 26 (5.8) |
| Other* | 33 (7.4) |
| How do you rate the support you obtained from your husband? | |
| Unsupportive | 59 (13.2) |
| Actively supportive | 266 (59.4) |
| Supportive on request | 93 (20.8) |
| Not applicable (no husband) | 30 (6.7) |
| Type of support at home** | |
| No support | 58 (12.9) |
| Baby care | 224 (50.0) |
| Staying with baby at home | 248 (55.4) |
| Household activities | 150 (33.5) |
| Other | 5 (1.1) |
| If no support at home, how you manage your child with work? | |
| Child minded at home by domestic worker | 149 (33.3) |
| Day care (outside home) | 24 (5.4) |
| Child taken to mother's work | 122 (27.2) |
| Other | 4 (0.9) |

*other: neighbours, extended family members

** possible to give more than one answer

## 3.4 Work-related factors affecting EBF

Four-hundred and forty participants identified a number of workplace factors that affected the continuation of EBF including receiving support from organizations, managers and co-workers, as well as availability of time and physical environment.

**3.4.1. Organizational support.** In responding to the organizational support related questions, 277 (63.0%) participants agreed/strongly agreed that they wished they had enough maternity leave before going back to work. Three-hundred and fourteen (70.7%) participants disagreed/strongly disagreed that they had policies about breastfeeding in their workplace. One hundred and ninety-six (44.5%) participants strongly disagreed and a further 191 (43.4%) disagreed that they had access to an area at work, specifically designated for breastfeeding. From the participant mothers, 313 (71.1%) reported that their employment would not be at risk if they breastfeed in their workplace. A total of 306 (69.6%) participants disagreed/strongly disagreed that their opportunities for job advancement would be limited if they breastfeed at work (Table 4).

**3.4.2. Managers' support.** Three hundred and twenty-three (73.4%) women disagreed/strongly disagreed that they had support from their managers to breastfeed at work. From the study participants, 293 (66.6%) mothers disagreed/strongly disagreed with the statement 'my

**Table 4. Work-related factors affecting EBF (n = 440).**

| Variables | n (%) |
|---|---|
| **Organizational support** | |
| I would have enough (paid or unpaid) maternity leave to get breastfeeding started before going back to work. | |
| Strongly disagree/Disagree | 163 (37.0) |
| Strongly agree/Agree | 277 (63.0) |
| My company has written policies for employees that breastfeed or express breast milk. | |
| Strongly disagree/Disagree | 314 (71.4) |
| Strongly agree/Agree | 126 (28.6) |
| I would feel comfortable asking for space to breastfeed or express breast milk at work | |
| Strongly disagree/Disagree | 357 (81.1) |
| Strongly agree/Agree | 83 (18.9) |
| I'm certain there is a place I could go to breastfeed or express breast milk at work. | |
| Strongly disagree/Disagree | 387 (88.0) |
| Strongly agree/Agree | 53 (12.0) |
| There is someone at work that would help me plan for breastfeed or express breast milk | |
| Strongly disagree/Disagree | 351 (79.8) |
| Strongly agree/Agree | 89 (20.2) |
| My job could be at risk (e.g. lose my job) if I breastfed or express breast milk at work | |
| Strongly agree/Agree | 316 (71.8) |
| Strongly disagree/Disagree | 124 (28.2) |
| My opportunities for job advancement would be limited if I breastfed/express breast milk at work | |
| Strongly agree/Agree | 306 (69.5) |
| Strongly disagree/Disagree | 134 (30.5) |
| **Managers support** | |
| My manager would support me breastfeeding or expressing breast milk at work | |
| Strongly disagree/Disagree | 320 (72.7) |
| Strongly agree/Agree | 120 (27.3) |
| My manager would think I couldn't finish all my work if I ask for a break for breastfeeding | |
| Strongly disagree/Disagree | 293 (66.6) |
| Strongly agree/Agree | 147 (33.4) |
| I would feel comfortable speaking with my manager about breastfeeding | |
| Strongly disagree/Disagree | 282 (64.1) |
| Strongly agree/Agree | 158 (35.9) |
| My manager would make sure my job is replaced if I need a break for breastfeeding or expressing breast milk | |

*(Continued)*

**Table 4.** (Continued)

| Variables | n (%) |
|---|---|
| Strongly disagree/Disagree | 199 (45.2) |
| Strongly agree/Agree | 241 (54.8) |
| My manager would change my work schedule to allow me time for breastfeeding or expressing breast milk | |
| Strongly disagree/Disagree | 319 (72.5) |
| Strongly agree/Agree | 121 (27.5) |
| My manager would help me deal with my workload to breastfeed/express breast milk | |
| Strongly disagree/Disagree | 316 (71.8) |
| Strongly agree/Agree | 124 (28.2) |
| **Co-workers' support** | |
| I would feel comfortable speaking with my co-workers about breastfeeding | |
| Strongly disagree/Disagree | 243 (55.2) |
| Strongly agree/Agree | 197 (44.8) |
| My co-workers would change their break times so that I could breastfeed/express breast milk | |
| Strongly disagree/Disagree | 188 (42.7) |
| Strongly agree/Agree | 252 (57.3) |
| My co-workers would replace my job duties if I needed time for breastfeeding or expressing breast milk. | |
| Strongly disagree/Disagree | 196 (44.5) |
| Strongly agree/Agree | 244 (55.5) |
| **Time related variables and Physical environment** | |
| My breaks are frequent enough for breastfeeding or expressing breast milk. | |
| Strongly disagree/Disagree | 358 (81.4) |
| Strongly agree/Agree | 82 (18.6) |
| I could adjust my break schedule in order to breastfeed or express breast milk. | |
| Strongly disagree/Disagree | 314 (71.4) |
| Strongly agree/Agree | 126 (28.6) |
| I could buy or borrow the equipment I would need for expressing breast milk. | |
| No | 356 (80.9) |
| Yes | 84 (19.1) |
| My company would supply the equipment I need for expressing breast milk at work. | |
| No | 417 (94.8) |
| Yes | 23 (5.1) |
| There is a company-designated place for women to breastfeed or express milk | |
| No | 440 (100) |
| Yes | 0 (0) |

manager would think I couldn't finish my work if I needed break for breastfeeding'. Two hundred and eighty-two (64.1%) mothers reported that they did not feel comfortable speaking about breastfeeding with managers. When talking about flexibility of the managers in supporting breastfeeding mothers, 241 (54.8%) participants reported that their managers want to make sure another person is available to undertake the work when the mothers needed time for breastfeeding. Ninety-one (20.7%) agreed and a further 30 (6.7%) strongly agreed, that their managers allowed them to change their work schedule for breastfeeding. However, 316 (71.8%) participants disclosed that their managers did not help them to manage their workload (Table 4).

**3.4.3. Co-workers support.** From the participant mothers, 243 (55.3%) did not feel comfortable when speaking with co-workers about breastfeeding. However, 243 (55.3%) agreed/ strongly agreed that their co-workers helped them by changing their break time to allow for breastfeeding or expressing breast milk. Similarly, 244 (55.4%) participants agreed/strongly agreed that their co-workers undertook their job to allow them time to breastfeed or express breast milk (Table 4).

**3.4.4. Time and physical environment.** From the study participants, 80 (18.1%) agreed/ strongly agreed that they had frequent enough breaks for breastfeeding or expressing breast milk. A total of 126 (28.6%) agreed/strongly agreed that they could adjust their schedule to make time for breastfeeding. Whereas, when talking about accessibility of equipment for breast milk expression, only 84 (19.1%) reported that they could buy or borrow equipment for expressing breast milk. All (100%) participants reported that none of the companies they work for had a designated place for women to breastfeed or express milk during the work day (Table 4).

## 3.5 EBF and associated factors among employed women

Two-hundred and fifty-four (56.6%) participants reported that they exclusively breastfed their infants until six months. The main reason for 46.4% of participants who did not adhere to EBF was returning to work before six months.

Of the variables used in the univariable logistic regression, only six variables had a p-value of <0.1 and were used in the multivariable logistic regression (Table 5). Mothers who had family support were two times more likely to continue EBF, compared to those who did not have family support (AOR = 2.1, 95%, CI 1.2–3.6; P = 0.005). Similarly, mothers who agreed/ strongly agreed of having frequent enough breaks were 2.6 times more likely to EBF than those who disagreed/strongly disagreed (AOR = 2.6, 95% CI, 1.4–4.8; P = 0.002). When the mothers could buy or borrow equipment needed for expressing breast milk, they were 1.6 times more likely to continue EBF compared to those who could not (AOR = 1.7, 95% CI, 1.0–3.0; P = 0.033). Table 5 mainly presented the adjusted odds ratio of the variables that show statistical association. However, the crude odds ratio of some important variables is included on Table 5.

## 4. Discussion

In this study more than half of the employed women reported EBF and the majority were aware of the benefits of EBF for infants and mothers and reported family support. While more than half of the participants agreed/strongly agreed that their co-workers helped them to change their work schedule, around three quarters of the participants disagreed/strongly disagreed that they had support from their managers. All participants reported that there is no designated place for women to breastfeed or express milk in their workplace. Over half of the study participants reported EBF at six months. Factors associated with EBF were having family

**Table 5. Logistic regression for work-related predictors of EBF among employed women (n = 440).**

| Variables | n | Crude OR (95% CI); p values | Adjusted OR (95% CI); p values |
|---|---|---|---|
| Age of mother (years) | | | NI |
| 18–29 | 265 | Ref. | |
| 30 or more | 175 | 1.2 (0.8–1.9); 0.263 | |
| Marital status | | | NI |
| Partnered | 373 | Ref. | |
| Unpartnered (single, divorced, widowed) | 67 | 1.2 (0.7–2.1); 0.439 | |
| Educational status | | | NI |
| Secondary or less | 103 | Ref. | |
| Diploma or more | 337 | 1.3 (0.7–2.2); 0.304 | |
| Monthly salary, (USD) | | | NI |
| 76 or less | 200 | Ref. | |
| Greater than 76 | 240 | 0.8 (0.5–1.3); 0.518 | |
| Number of children | | | NI |
| One | 125 | Ref. | |
| Two or more | 315 | 0.7 (0.4–1.1); 0.166 | |
| Awareness about breast milk expression | | | |
| No | 151 | Ref. | Ref |
| Yes | 289 | 1.5 (1.0–2.3); 0.046 | 1.1 (0.7–1.7); 0.469 |
| Family support to continue EBF following return to paid employment | | | |
| No | 77 | Ref. | Ref |
| Yes | 363 | 2.3 (1.4–3.9); 0.001 | 2.1 (1.2–3.6); 0.009 |
| I would have enough maternity leave (paid and/or unpaid) to get BF started before going back to work | | | |
| Strongly disagree/Disagree | 163 | Ref. | NI |
| Strongly agree/Agree | 277 | 1.2(0.7–1.9); 0.351 | |
| I'm certain there is a place I could go to breastfeed or express breast milk at work | | | |
| Strongly disagree/Disagree | 387 | Ref. | Ref |
| Strongly agree/Agree | 53 | 0.4 (0.2–1.0); 0.081 | 0.7 (0.4–1.5); 0.455 |
| I would feel comfortable asking for accommodation for breastfeeding or express breast milk at work. | | | |
| Strongly disagree/Disagree | 357 | Ref. | Ref. |
| Strongly agree/Agree | 83 | 0.5 (0.2–1.1); 0.099 | 0.6 (0.3–1.2); 0.182 |
| My manager would change my work schedule to allow me time to breastfeed | | | |
| Strongly disagree/Disagree | 320 | Ref. | Ref. |
| Strongly agree/Agree | 120 | 0.4 (0.2–1.0); 0.062 | 0.7 (0.4–1.2); 0.282 |
| My breaks are frequent enough for breastfeeding or expressing breast milk. | | | |
| Strongly disagree/Disagree | 358 | Ref. | Ref. |
| Strongly agree/Agree | 82 | 2.5 (1.1–5.6); 0.018 | 2.6 (1.4–4.8); 0.002 |
| I could buy or borrow the equipment I would need for expressing breast milk. | | | |
| No | 356 | Ref. | Ref |
| Yes | 84 | 2.0 (1.1–3.8); 0.022 | 1.7 (1.0–3.0); 0.033 |

NI: Not included

support, having frequent breaks at work and the possibility of buying or borrowing equipment for expressing breast milk.

The prevalence of EBF in our study (56.4%) was higher compared to other studies conducted in other areas of Ethiopia including Gondar in 2015 (20.9%), Dukem in 2015 (24.3%) and Fafan in 2016 (24.8%) [2, 7, 8]. The reason for the increase in EBF might be a consequence of recent improvements in maternity leave in Ethiopia. Besides, women can use their annual leave after they finish their four months maternity leave which assists employed women to extend the duration of EBF at least by one month. Our prevalence of EBF was also higher compared to other reported prevalence in other areas of Africa such as Ghana (10.3%) and Egypt (14.1%) [18, 19]. The difference might be due to the difference in leave entitlements, or local cultural practices and social expectations.

Awareness of mothers about the benefits of EBF is crucial for its continuation. In this study 98.2% participants believe that EBF has benefits. A similar finding was obtained in a study conducted in Fafan (Somali region of Ethiopia) in which all participants were aware of the benefits of EBF [2]. Studies conducted in Ghana and South Jordan also found that 99% and 99.3% mothers were aware of the benefits of EBF, respectively [18, 20]. However, in studies conducted in Gondar, Ethiopia and Nigeria, 80% and 77.5% participants acknowledged the benefits of EBF, respectively [7, 21]. These figures suggested that employed women have good awareness of the benefits of EBF which could motivate them to continue EBF until six months, even after they have returned to work [7].

In this study, having family support was positively associated with continuation of EBF. Common supports mothers obtained at home include baby care, staying with baby at home while she was at work, helping with cooking and other household activities. A similar finding was obtained in a study conducted among working women in Indonesia [22]. The authors of the Indonesian study found that mothers who had family support were two times more likely than those who did not have the support to exclusively breastfeed [22]. However, a study conducted in Gondar, Ethiopia showed that mothers who had no family support were more likely to exclusively breastfeed as compared to those who had family support [7]. This was an unexpected finding not supported by the literature. Women who have support at home could spend more time with their infants which helped the children get adequate breastmilk. Therefore, having family support encouraged women to continue EBF after they returned to paid employment.

Participants who had frequent breaks at work were more likely to continue EBF. In this study mothers who agreed/strongly agreed that they had frequent enough breaks at work were more likely to continue EBF as compared to those who disagreed/strongly disagreed. Similarly, in the study conducted in Dukem, central Ethiopia [8], mothers who had no break time at work were more likely to discontinue EBF as compared to those who had breaks. This would be because having sufficient breaks might encourage women to continue EBF after they returned to work [23]. When employed women have breaks they could either go home to breastfeed their infants if their home is nearby or ask someone to bring their infants to work for breastfeeding.

Participants who could buy or borrow equipment they need for expressing breast milk, were more likely to continue EBF as compared to those who could not afford it. This means, if mothers have access to equipment to express their breast milk, they can exclusively breastfeed for longer. However, there is no existing literature to compare with this finding. Therefore, this could be a new finding in this study.

This study has limitations. Firstly, self-administered questionnaire was used to collect the data which makes it difficult for women to seek clarification for any question. Secondly, employed women with children up to two years were included in the study which could lead

to recall bias of the exact duration of EBF. Thirdly, the study participants were government employees. Therefore, the findings would not represent women employed in private organizations. Further research with a focus on women employed in private organizations is recommended. Lastly, we used convenience sampling in northern Ethiopia and the findings might not be generalizable to all employed women in Ethiopia.

## 5. Conclusion

Almost all participants were aware of the benefits of breastfeeding. Our findings show increased percentage of EBF among employed women in Ethiopia compared to previous studies [2, 7, 8]. We tend to think that this might be partly contributed by the recent increase in maternity leave in Ethiopia because many of the children in this study were born after July 2018, when the policy changed. Although other factors play roles in EBF, based on the most common reason for EBF cessation in this study, we believe that six-month's maternity leave might be the most helpful solution for working women to be able to exclusively breastfeed their baby. In the absence of six-month maternity leave, the role of families and managers' role is of importance in continuing EBF is critical.

## Supporting information

**S1 Data.**
(SAV)

**S1 Table. Logistic regression for predictors of EBF among employed women in North Ethiopia. (n = 440).**
(DOCX)

## Acknowledgments

The authors would like to acknowledge the participant mothers for sharing their experiences by responding to the questionnaire. We would also like to thank Dr Lucy Busija and Mr Ian Hunt of Monash Statistical Consulting Service for their statistical advice.

## Author Contributions

**Conceptualization:** Kahsu Gebrekidan, Helen Hall, Virginia Plummer, Ensieh Fooladi.

**Formal analysis:** Kahsu Gebrekidan.

**Investigation:** Kahsu Gebrekidan.

**Methodology:** Kahsu Gebrekidan.

**Software:** Kahsu Gebrekidan.

**Supervision:** Helen Hall, Virginia Plummer, Ensieh Fooladi.

**Writing – original draft:** Kahsu Gebrekidan.

**Writing – review & editing:** Kahsu Gebrekidan, Helen Hall, Virginia Plummer, Ensieh Fooladi.

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
