## [Decision Letter · Decision Letter 0]

31 Dec 2020

PONE-D-20-37978

Exclusive breastfeeding continuation and associated factors among employed women in North Ethiopia: A cross-sectional study

PLOS ONE

Dear Dr. Gebrekidan,

Thank you for submitting your manuscript to PLOS ONE. After careful consideration, we feel that it has merit but does not fully meet PLOS ONE’s publication criteria as it currently stands. Therefore, we invite you to submit a revised version of the manuscript that addresses the points raised during the review process.

An expert in the field handled your manuscript, and we are grateful for their time and contributions. Although interest was found in your study, several major concerns arose that require your attention. Please address ALL of the reviewer's comments in your revised manuscript.

We look forward to receiving your revised manuscript.

Kind regards,

Frank T. Spradley

Academic Editor

PLOS ONE

2. Please include additional information regarding the survey or questionnaire used in the study and ensure that you have provided sufficient details that others could replicate the analyses. For instance, if you developed a questionnaire as part of this study and it is not under a copyright more restrictive than CC-BY, please include a copy, in both the original language and English, as Supporting Information, or include a citation if it has been published previously.

3. In the Methods, please discuss whether and how the questionnaire was validated.

4. In statistical methods, please refer to any post-hoc corrections to correct for multiple comparisons during your statistical analyses. If these were not performed please justify the reasons. Please refer to our statistical reporting guidelines for assistance (https://journals.plos.org/plosone/s/submission-guidelines.#loc-statistical-reporting).

Reviewers' comments:

Reviewer's Responses to Questions

**Comments to the Author**

1. Is the manuscript technically sound, and do the data support the conclusions?

Reviewer #1: Partly

2. Has the statistical analysis been performed appropriately and rigorously? 

Reviewer #1: I Don't Know

3. Have the authors made all data underlying the findings in their manuscript fully available?

Reviewer #1: Yes

4. Is the manuscript presented in an intelligible fashion and written in standard English?

Reviewer #1: Yes

5. Review Comments to the Author

Reviewer #1: This is a very important and interesting study so I'm keen to see it published. I am thanking the authors for their hard work and hope that my comments below will be useful for improving the manuscript.

Major comments and/or request for clarifications:

1. P.7, line 125-129: Sample size. This manuscript was looking at the prevalence among EBF among employed women. In this case, I think the authors should have been using an estimated prevalence of EBF among employed women (from previous studies) as the reference to calculate their sample, instead of prevalence of employed women in the community. This also brought further comment to the result and discussion. The ORs that they got from multivariable analysis ranged from 0.6-2.6, so for some variables, and I am worried that their sample may not be enough. If this is a limitation, it may be helpful to mention it in the discussion. Otherwise, I would be happy to receive any clarification.

2. P7, line 130-141. Statistical analysis. I believe that the authors have carefully considered the statistical approach to measure effect size. However, given the high prevalence of outcome in this study population, I wonder why the authors chose logistic regression. I am aware that OR is widely used to approximate RR as the ‘true’ effect size in studies that investigate associated factors. However, ORs produced by logistic regression in a sample with prevalent outcome tend to be overestimated. Apologies if I missed it, but it was not mentioned in the discussion and I was struggling to find the authors’ justification for not using other statistical approach such as modified Poisson regression to estimate effect size. I hope these can be useful references (Barros and Hirakata, 2003, https://pubmed.ncbi.nlm.nih.gov/14567763/ and Zou, 2004, https://academic.oup.com/aje/article/159/7/702/71883).

3. Coherence between introduction and conclusion. In line 309-310, the authors stated, “Our findings show a reassuring increased percentage of EBF among employed women which might be due to a recent increase in maternity leave in Ethiopia”. According to the introduction in line 78, policy change of maternity leave in Ethiopia (increased length from three to four months) took place in 2019. This study was done later in December 2018 – January 2019 (line 89) and the study participants were women whose child was six-month-old up to two-year-old. If this study was to assess EBF continuation and associated factors among employed women since the introduction of increased maternal leave (line 80), then to my understanding, this timeline could not support authors’ conclusion.

4. Table 2. Reasons for not continuing EBF. I wonder if the participants can only pick one. Out of curiosity, I would love to hear further explanation on the situation where someone had two or more reasons.

5. Table 2. Some women cannot breastfeed because of medical conditions. I did not see medical or health-related reasons being mentioned or discussed in this study. Was this condition an exclusion criterion, was it not asked, or none of the participants experienced it?

6. Table 3. In row 2, domestic worker was listed as part of family support, but in row 5, it was asked as an option for participants who did not have support at home. This is a bit confusing and I hope the authors can clarify.

7. Table 4. Question “I would have enough (paid or unpaid) maternity leave to get breastfeeding started before going back to work”. In line 198, it meant the participants felt that they had enough maternity leave. I would rephrase the question so that it would be clearer. I initially was not sure whether that statement meant the participants had enough maternity leave or it was something that they wished they had/desired in the future.

8. Table 5. I wonder how the authors selected the variables to include in their multivariable analysis. I read that only complete question was included, but is there any other consideration? What about other important variables, e.g. having enough maternity leave? In my opinion, this variable is highly relevant to the study objective. However, I did not manage to find the reason it was not included while some other work-related variables (e.g. certainty about place to express breastmilk at work) were.

9. Table 5. I observe a major flip between all the crude ORs and adjusted ORs. Was the reference category for the crude ORs different from that for adjusted ORs?

10. Line 313. The authors stated, “Maternal leave for six months is the best solution for women to adhere to EBF”. As much as I personally want this policy to come true, I struggled to find which part of the findings that supported this conclusion.

Minor comments:

1. P2, line 26 and elsewhere: terminology. To my knowledge, the official term is “convenience sampling” instead of “convenient sampling” (Sedgwick, 2013, https://www.bmj.com/content/347/bmj.f6304).

2. P2, line 30 and elsewhere: terminology; multivariable instead multivariate (and likewise, P7 line 138 and elsewhere: univariable instead of univariate). To my understanding, multivariate means multiple outcomes (dependent variables) and multivariable means multiple independent variables (Hidalgo and Goodman, 2013, https://www.ncbi.nlm.nih.gov/pmc/articles/PMC3518362/). I am aware that there has been a debate on this terminology over the years, so I am open for other arguments/insights either from the authors, other reviewers, or the editors.

3. P4, line 47-49: citations. I would recommend authors to cite the original document i.e. WHO/UNICEF policy or statement, rather than cite other research articles that cite it.

4. Line 234. It may be helpful for readers if the authors indicate which table they are referring to.

5. Line 244 and 296. I believe the authors meant EBF instead of EBE.

6. PLOS authors have the option to publish the peer review history of their article (what does this mean?). If published, this will include your full peer review and any attached files.

Reviewer #1: **Yes: **Lhuri D. Rahmartani

---

## [Author Response · Author response to Decision Letter 0]

3 Mar 2021

PONE-D-20-37978

Exclusive breastfeeding continuation and associated factors among employed women in North Ethiopia: A cross-sectional study

Dear Dr Frank T. Spradley,

The Editor PLOS ONE

Thank you for considering our manuscript for publication in the “PLOS ONE” Journal. 

We have addressed the editor and reviewers’ helpful comments point by point as follows. 

Kind regards,

Kahsu Gebrekidan, on behalf of all authors

 

Reviewer #1: 

This is a very important and interesting study so I'm keen to see it published. I am thanking the authors for their hard work and hope that my comments below will be useful for improving the manuscript.

• Response: Thank you very much.

Major comments and/or request for clarifications:

1. P.7, line 125-129: Sample size. This manuscript was looking at the prevalence among EBF among employed women. In this case, I think the authors should have been using an estimated prevalence of EBF among employed women (from previous studies) as the reference to calculate their sample, instead of prevalence of employed women in the community. 

• Response: Thank you for your comment, it was a typographical error now corrected on page 7, line 127. The sample size was calculated using estimated prevalence of EBF among employed women obtained from a study conducted in Gondar, Ethiopia (ref. 7). 

• This also brought further comment to the result and discussion. The ORs that they got from multivariable analysis ranged from 0.6-2.6, so for some variables, and I am worried that their sample may not be enough. If this is a limitation, it may be helpful to mention it in the discussion. Otherwise, I would be happy to receive any clarification.

• Response: Thank you for your concern. Please note that the CI of 0.6- 2.6 is not considered wide. Our sample size was for the logistic regression was based on the rule of thumb for sample size in regression analysis is that for regression equations using six or more predictors, an absolute minimum of 10 participants per predictor variable is appropriate. (ref: Harris, R. J. (1985). A primer of multivariate statistics (2nd ed.). New York: Academic Press).

2. P7, line 130-141. Statistical analysis. I believe that the authors have carefully considered the statistical approach to measure effect size. However, given the high prevalence of outcome in this study population, I wonder why the authors chose logistic regression. I am aware that OR is widely used to approximate RR as the ‘true’ effect size in studies that investigate associated factors. However, ORs produced by logistic regression in a sample with prevalent outcome tend to be overestimated. Apologies if I missed it, but it was not mentioned in the discussion and I was struggling to find the authors’ justification for not using other statistical approach such as modified Poisson regression to estimate effect size. I hope these can be useful references (Barros and Hirakata, 2003, https://pubmed.ncbi.nlm.nih.gov/14567763/and Zou,2004, https://academic.oup.com/aje/article/159/7/702/71883).

• Response: Thank you very much for recommending the helpful references and for the suggestion. While your point is valid, however, logistic regression analysis for binary outcomes is frequently used in epidemiological studies. 

3. Coherence between introduction and conclusion. In line 309-310, the authors stated, “Our findings show a reassuring increased percentage of EBF among employed women which might be due to a recent increase in maternity leave in Ethiopia”. According to the introduction in line 78, policy change of maternity leave in Ethiopia (increased length from three to four months) took place in2018. This study was done later in December 2018 – January 2019 (line 89) and the study participants were women whose child was six-month-old up to two-year-old. If this study was to assess EBF continuation and associated factors among employed women since the introduction of increased maternal leave (line 80), then to my understanding, this timeline could not support authors’ conclusion.

• Response: Thank you for your comment. The increment of the maternity leave was started on 1st July 2018 and line 78 has been corrected to reflect this.

4. Table 2. Reasons for not continuing EBF. I wonder if the participants can only pick one. Out of curiosity, I would love to hear further explanation on the situation where someone had two or more reasons.

• Response: Thank you for your comment. On the questionnaire, it was stated that the participants could provide more than one answer, an alternative there was “others (specify)”. As a result, 16 of 210 participants responded more than one answer.

5. Table 2. Some women cannot breastfeed because of medical conditions. I did not see medical or health-related reasons being mentioned or discussed in this study. Was this condition an exclusion criterion, was it not asked, or none of the participants experienced it?

• Response: We agree that medical condition could be a factor for EBF discontinuation. In our study it was possible to give a related answer under the response “if others (please specify)” but none of the participants experienced it. 

6. Table 3. In row 2, domestic worker was listed as part of family support, but in row 5, it was asked as an option for participants who did not have support at home. This is a bit confusing and I hope the authors can clarify.

• Response: Thank you for your comment. Some participants mentioned that the domestic workers were part of their extended family members and responded under family support. This category was added during data coding because many participants responded this under the category “if others (please specify)”. For other participants the domestic workers were not considered as part of their extended family. 

7. Table 4. Question “I would have enough (paid or unpaid) maternity leave to get breastfeeding started before going back to work”. In line 198, it meant the participants felt that they had enough maternity leave. I would rephrase the question so that it would be clearer. I initially was not sure whether that statement meant the participants had enough maternity leave or it was something that they wished they had/desired in the future.

• Response: Thank you for your comment. The statement has been rephrased to include the recommended changes on page 12 line 199

8. Table 5. I wonder how the authors selected the variables to include in their multivariable analysis. I read that only complete question was included, but is there any other consideration? What about other important variables, e.g. having enough maternity leave? In my opinion, this variable is highly relevant to the study objective. However, I did not manage to find the reason it was not included while some other work-related variables (e.g. certainty about place to express breastmilk at work) were.

• Response: Thank you for your comment. Initially all completed variables (n=440) were included in the univariable logistic regression analysis. As a result, all the work-related variables were included. Some variables from part 2: breastfeeding knowledge and practice and part 3: family support to EBF were excluded. Variables with P-value <0.1 in the univariable logistic regression were included in the multivariable logistic regression.

9. Table 5. I observe a major flip between all the crude ORs and adjusted ORs. Was the reference category for the crude ORs different from that for adjusted ORs?

• Response: Thank you for your feedback, the reference category for the COR has been reversed, now it is amended (Table 5 page 17)

10. Line 313. The authors stated, “Maternal leave for six months is the best solution for women to adhere to EBF”. As much as I personally want this policy to come true, I struggled to find which part of the findings that supported this conclusion.

• Response: Thank you for your comment. In our study the main reason many mothers discontinue EBF was returning to work before six months (page 10 line 170). That is why maternal leave for six months is the best solution for women to adhere to EBF.

Minor comments:

1. P2, line 26 and elsewhere: terminology. To my knowledge, the official term is “convenience sampling” instead of “convenient sampling” (Sedgwick, 2013, https://www.bmj.com/content/347/bmj.f6304).

• Response: Thank you for your feedback, all references to convenient sampling have been changed to convenience sampling on pages 2, 6 and 20

2. P2, line 30 and elsewhere: terminology; multivariable instead multivariate (and likewise, P7 line 138 and elsewhere: univariable instead of univariate). To my understanding, multivariate means multiple outcomes (dependent variables) and multivariable means multiple independent variables (Hidalgo and Goodman, 2013, https://www.ncbi.nlm.nih.gov/pmc/articles/PMC3518362/). I am aware that there has been a debate on this terminology over the years, so I am open for other arguments/insights either from the authors, other reviewers, or the editors.

• Response: Thank you for your comment. The recommended changes are done on pages 2, 7, 16: univariate changed to univariable and multivariate changed to multivariable.

---

## [Decision Letter · Decision Letter 1]

7 Apr 2021

PONE-D-20-37978R1

Exclusive breastfeeding continuation and associated factors among employed women in North Ethiopia: A cross-sectional study

PLOS ONE

Dear Dr. Gebrekidan,

Thank you for submitting your manuscript to PLOS ONE. After careful consideration, we feel that it has merit but does not fully meet PLOS ONE’s publication criteria as it currently stands. Therefore, we invite you to submit a revised version of the manuscript that addresses the points raised during the review process.

We look forward to receiving your revised manuscript.

Kind regards,

Frank T. Spradley

Academic Editor

PLOS ONE

Reviewers' comments:

Reviewer's Responses to Questions

**Comments to the Author**

1. If the authors have adequately addressed your comments raised in a previous round of review and you feel that this manuscript is now acceptable for publication, you may indicate that here to bypass the “Comments to the Author” section, enter your conflict of interest statement in the “Confidential to Editor” section, and submit your "Accept" recommendation.

Reviewer #1: All comments have been addressed

2. Is the manuscript technically sound, and do the data support the conclusions?

Reviewer #1: Yes

3. Has the statistical analysis been performed appropriately and rigorously? 

Reviewer #1: I Don't Know

4. Have the authors made all data underlying the findings in their manuscript fully available?

Reviewer #1: Yes

5. Is the manuscript presented in an intelligible fashion and written in standard English?

Reviewer #1: Yes

6. Review Comments to the Author

Reviewer #1: Dear Authors,

Many thanks for addressing my comments and for your hard work revising the draft. I only have a few further responses as listed below.

Major comments:

1. Point 3 in previous review (now line 310). While the policy change has been corrected from 2019 to July 2018, I would still partly disagree with your conclusion that the increase in EBF might be due to the recent increase of maternity leave length from three to four months. This is because the mean age of the child was 12 months which means that most of the child was born before July 2018. I suggest that if you want to keep this conclusion in line 310, please rephrase it to “…our findings show increased percentage of EBF among employed women in Ethiopia compared to (previous studies [insert ref]). We tend to think that this might be partly contributed by the recent increase in maternity leave in Ethiopia because many of the children in this study were born after July 2018, when the policy changed. However, most of them were also born before July 2018, so we do not have enough evidence to conclude that. Moreover, this increase of EBF may also be caused by other factors especially related to study selection. Hence, while we support this policy change, our assumption on the benefit of extended maternity leave will need further investigation.”

2. Points 8 in my initial review (Table 5). Thank you for explaining how you selected the variables in the final model. I think this explanation on variable selection should be more clearly mentioned in methods section (line 131). If all complete variables were initially included in the univariable logistic regression as you said, then I would expect to see crude OR for all variables that had n=440, although it will mean a longer table. Consequently, if “having enough maternity leave” is an important factor, I believe it should be shown in Table 5. Even if it’s not statistically significant, at least its crude OR should be shown, because it had n=440. If you do not wish to show crude ORs for all variables, then you may need to explain further why some variables were shown in Table 5 and some were not. That way readers know what happened with all variables in previous tables, especially the important ones, like the maternity leave, that you mentioned in the conclusion.

3. Points 10 in my initial review (now line 314). I understand your reasoning, but I don’t think this study was designed to reach that conclusion. I would suggest something less intense like, “Although other factors play roles in EBF, based on the most common reason for EBF cessation in this study, we believe that six-month maternity leave might be the most helpful solution for working women to be able to exclusively breastfeed their baby.”

Minor comment:

1. I believe ‘maternity leave’ is the official and more widely known term. You have mostly used it in the previous draft, so I wonder why you changed it to ‘maternal leave’ in this version.

7. PLOS authors have the option to publish the peer review history of their article (what does this mean?). If published, this will include your full peer review and any attached files.

Reviewer #1: **Yes: **Lhuri D. Rahmartani

---

## [Author Response · Author response to Decision Letter 1]

14 Apr 2021

Reviewer #1: 

Major comments:

1. Point 3 in previous review (now line 310). While the policy change has been corrected from 2019 to July 2018, I would still partly disagree with your conclusion that the increase in EBF might be due to the recent increase of maternity leave length from three to four months. This is because the mean age of the child was 12 months which means that most of the child was born before July 2018. I suggest that if you want to keep this conclusion in line 310, please rephrase it to “…our findings show increased percentage of EBF among employed women in Ethiopia compared to (previous studies [insert ref]). We tend to think that this might be partly contributed by the recent increase in maternity leave in Ethiopia because many of the children in this study were born after July 2018, when the policy changed. However, most of them were also born before July 2018, so we do not have enough evidence to conclude that. Moreover, this increase of EBF may also be caused by other factors especially related to study selection. Hence, while we support this policy change, our assumption on the benefit of extended maternity leave will need further investigation.”

• Response: Thank you for your comment, the conclusion is revised to include suggested changes on page 21, lines 311-326

2. Points 8 in my initial review (Table 5). Thank you for explaining how you selected the variables in the final model. I think this explanation on variable selection should be more clearly mentioned in methods section (line 131). If all complete variables were initially included in the univariable logistic regression as you said, then I would expect to see crude OR for all variables that had n=440, although it will mean a longer table. Consequently, if “having enough maternity leave” is an important factor, I believe it should be shown in Table 5. Even if it’s not statistically significant, at least its crude OR should be shown, because it had n=440. If you do not wish to show crude ORs for all variables, then you may need to explain further why some variables were shown in Table 5 and some were not. That way readers know what happened with all variables in previous tables, especially the important ones, like the maternity leave, that you mentioned in the conclusion.

• Response: Thank you for your comment, we have included the crude OR of having enough maternity leave on Table 5. However, it is difficult to include the crude OR of all the variables because it would be a very lengthy table (4 pages). For further details the crude odds ratio for all variables is attached as a supporting material. 

3. Points 10 in my initial review (now line 314). I understand your reasoning, but I don’t think this study was designed to reach that conclusion. I would suggest something less intense like, “Although other factors play roles in EBF, based on the most common reason for EBF cessation in this study, we believe that six-month maternity leave might be the most helpful solution for working women to be able to exclusively breastfeed their baby.”

• Response: Thank you for your comment, the conclusion amended to include the recommended change on page 21, lines 311-326

Minor comment:

1. I believe ‘maternity leave’ is the official and more widely known term. You have mostly used it in the previous draft, so I wonder why you changed it to ‘maternal leave’ in this version.

• Response: Thank you for your comment, maternal leave is changed to maternity leave throughout the manuscript

---

## [Decision Letter · Decision Letter 2]

7 May 2021

PONE-D-20-37978R2

Exclusive breastfeeding continuation and associated factors among employed women in North Ethiopia: A cross-sectional study

PLOS ONE

Dear Dr. Gebrekidan,

Thank you for submitting your manuscript to PLOS ONE. After careful consideration, we feel that it has merit but does not fully meet PLOS ONE’s publication criteria as it currently stands. Therefore, we invite you to submit a revised version of the manuscript that addresses the points raised during the review process.

It is requested that the authors include a biostatistician on this study to confirm the statistical methods used. Also, the authors should contact a copyeditor to proofread this manuscript before resubmission.

We look forward to receiving your revised manuscript.

Kind regards,

Frank T. Spradley

Academic Editor

PLOS ONE

Reviewers' comments:

Reviewer's Responses to Questions

**Comments to the Author**

1. If the authors have adequately addressed your comments raised in a previous round of review and you feel that this manuscript is now acceptable for publication, you may indicate that here to bypass the “Comments to the Author” section, enter your conflict of interest statement in the “Confidential to Editor” section, and submit your "Accept" recommendation.

Reviewer #1: All comments have been addressed

2. Is the manuscript technically sound, and do the data support the conclusions?

Reviewer #1: Partly

3. Has the statistical analysis been performed appropriately and rigorously? 

Reviewer #1: Yes

4. Have the authors made all data underlying the findings in their manuscript fully available?

Reviewer #1: Yes

5. Is the manuscript presented in an intelligible fashion and written in standard English?

Reviewer #1: Yes

6. Review Comments to the Author

Reviewer #1: I thank authors for quickly responding to my comments. I personally think this manuscript is now ready for publication. I just wish the editor would look at the reasoning of the use of logistic regression in this study, or ask reviewers with better understanding in statistics because I don't think I have enough expertise in this and I am still learning too, hence I might be wrong. I'm aware that logistic regression is widely used, even in population with prevalent outcome. In my understanding, logistic regression analysis in studies with prevalent outcome may overestimate effect size, so that is why I thought of other approaches like modified Poisson regression in the first place. I'm not saying the authors have to do the same thing/change their methods, because they did the regression properly. However in my opinion, if we end up using logistic regression in such studies, perhaps it's worth mentioning something in discussion section about the possibility of overestimated effect sizes. Feel free to disagree with me though.

I'm also not a native/highly proficient English speaker so I can't guarantee the grammatical correctness of this manuscript. While this is written in intelligible and standard English, it would be great if authors can do a final proofread. I spotted one minor typological error in line 249-250:

"....However, the crude odds ratio of some important variables if included on Table 5." I think it should be "..However, the crude odds ratio of some important variables is included on Table 5.", as I believe the authors meant 'is' not 'if'.

Other than that, authors have addressed my comments and revised the draft accordingly. I hope my comments are useful and I wish authors all the best with this publication. Thank you.

7. PLOS authors have the option to publish the peer review history of their article (what does this mean?). If published, this will include your full peer review and any attached files.

Reviewer #1: **Yes: **Lhuri D. Rahmartani

---

## [Author Response · Author response to Decision Letter 2]

12 May 2021

PONE-D-20-37978

Exclusive breastfeeding continuation and associated factors among employed women in North Ethiopia: A cross-sectional study

Dear Editor

PLOS ONE

Thank you for considering our manuscript for publication in the “PLOS ONE” Journal. 

We have addressed the editor and reviewers’ helpful comments point by point as follows. 

Kind regards,

Kahsu Gebrekidan, on behalf of all authors

 

Reviewer #1: 

Major comments:

1. It is requested that the authors include a biostatistician on this study to confirm the statistical methods used. 

• Response: Thank you for your comment, statisticians are included in the acknowledgement on page 22, line 324-325

2. The authors should contact a copyeditor to proofread this manuscript before resubmission.

• Response: Thank you for your comment, the whole manuscript is proofread by a professional editor.

---

## [Decision Letter · Decision Letter 3]

2 Jun 2021

PONE-D-20-37978R3

Exclusive breastfeeding continuation and associated factors among employed women in North Ethiopia: A cross-sectional study

PLOS ONE

Dear Dr. Gebrekidan,

Thank you for submitting your manuscript to PLOS ONE. After careful consideration, we feel that it has merit but does not fully meet PLOS ONE’s publication criteria as it currently stands. Therefore, we invite you to submit a revised version of the manuscript that addresses the points raised during the review process.

We look forward to receiving your revised manuscript.

Kind regards,

Frank T. Spradley

Academic Editor

PLOS ONE

Journal Requirements:

Reviewers' comments:

Reviewer's Responses to Questions

**Comments to the Author**

1. If the authors have adequately addressed your comments raised in a previous round of review and you feel that this manuscript is now acceptable for publication, you may indicate that here to bypass the “Comments to the Author” section, enter your conflict of interest statement in the “Confidential to Editor” section, and submit your "Accept" recommendation.

Reviewer #2: (No Response)

2. Is the manuscript technically sound, and do the data support the conclusions?

Reviewer #2: Yes

3. Has the statistical analysis been performed appropriately and rigorously? 

Reviewer #2: Yes

4. Have the authors made all data underlying the findings in their manuscript fully available?

Reviewer #2: Yes

5. Is the manuscript presented in an intelligible fashion and written in standard English?

Reviewer #2: Yes

6. Review Comments to the Author

Reviewer #2: A cross-sectional study was conducted to estimate the prevalence of Exclusive Breastfeeding (EBF) continuation and associated factors, collected via BESt questionnaire, among employed women in North Ethiopia. Logistic regression modeling was applied. Fifty-six percent of women continued EBF for six months or more. Having family support, frequent breaks at work and purchasing or borrowing breast expressing equipment were significant factors in the multivariate model for predicting EBF.

Minor revisions:

1- Line 126: Indicate the statistical testing method which achieves 80%.

2- Provide a 95% confidence interval for the proportion with the duration of EBF of 6 months or more.

3- To clarify the data in table 5, separate the OR and 95% CI from the p-values. Use two columns to summarize univariate results and another two columns to summarize multivariate results. Clearly label them univariate and multivariate results.

7. PLOS authors have the option to publish the peer review history of their article (what does this mean?). If published, this will include your full peer review and any attached files.

Reviewer #2: No

---

## [Author Response · Author response to Decision Letter 3]

5 Jun 2021

Reviewer #2: 

A cross-sectional study was conducted to estimate the prevalence of Exclusive Breastfeeding (EBF) continuation and associated factors, collected via BESt questionnaire, among employed women in North Ethiopia. Logistic regression modelling was applied. Fifty-six percent of women continued EBF for six months or more. Having family support, frequent breaks at work and purchasing or borrowing breast expressing equipment were significant factors in the multivariate model for predicting EBF.

• Response: Thank you, however, I am not clear where this paragraph belongs to?

Minor revisions:

1- Line 126: Indicate the statistical testing method which achieves 80%

• Response: Thank you for your feedback, we have mentioned it under data analysis 

2- Provide a 95% confidence interval for the proportion with the duration of EBF of 6 months or more.

• Response: Thank you, the proportion with the duration of EBF amended to include the recommended changes

3- To clarify the data in table 5, separate the OR and 95% CI from the p-values. Use two columns to summarize univariate results and another two columns to summarize multivariate results. Clearly label them univariate and multivariate results.

• Response: Thank you, table 5 Amended to include the recommended changes

---

## [Editor Report · Decision Letter 4]

15 Jun 2021

Exclusive breastfeeding continuation and associated factors among employed women in North Ethiopia: A cross-sectional study

PONE-D-20-37978R4

Dear Dr. Gebrekidan,

We’re pleased to inform you that your manuscript has been judged scientifically suitable for publication and will be formally accepted for publication once it meets all outstanding technical requirements.

Kind regards,

Frank T. Spradley

Academic Editor

PLOS ONE

---

## [Editor Report · Acceptance letter]

15 Jun 2021

PONE-D-20-37978R4 

Exclusive breastfeeding continuation and associated factors among employed women in North Ethiopia: A cross-sectional study 

Dear Dr. Gebrekidan:

I'm pleased to inform you that your manuscript has been deemed suitable for publication in PLOS ONE. Congratulations! Your manuscript is now with our production department. 

Kind regards, 

on behalf of

Dr. Frank T. Spradley 

Academic Editor

PLOS ONE